# Fertility and contraceptive dynamics amidst COVID-19: who is at greatest risk for unintended pregnancy among a cohort of adolescents and young adults in Nairobi, Kenya?

Shannon N Wood  ,[1] Meagan E Byrne,[1] Mary Thiongo,[2] Bianca Devoto,[1] Grace Wamue-Ngare,[3] Michele R Decker,[1] Peter Gichangi[2]

[1]Department of Population, Family and Reproductive Health, Johns Hopkins University Bloomberg School of Public Health, Baltimore, Maryland, USA
[2]International Centre for Reproductive Health Kenya, Mombasa, Kenya
[3]Department of Sociology, Gender and Development Studies, Kenyatta University, Nairobi, Kenya

**Correspondence to**
Dr Shannon N Wood;
swood@jhu.edu

## ABSTRACT

**Objectives** Among youth in Nairobi, we (1) characterised fertility and contraceptive use dynamics by gender; (2) estimated pregnancy prevalence over the pandemic; and (3) assessed factors associated with unintended pandemic pregnancy for young women.

**Design** Longitudinal analyses use cohort data collected at three timepoints prior to and during the COVID-19 pandemic: June to August 2019 (pre-pandemic), August to October 2020 (12-month follow-up) and April to May 2021 (18-month follow-up).

**Setting** Nairobi, Kenya.

**Participants** At initial cohort recruitment, eligible youth were aged 15–24 years, unmarried and residing in Nairobi for at least 1 year. Within-timepoint analyses were restricted to participants with survey data per round; trend and prospective analyses were restricted to those with complete data at all three timepoints (n=586 young men, n=589 young women).

**Primary and secondary outcome measures** Primary outcomes comprised fertility and contraceptive use for both genders, and pregnancy for young women. Unintended pandemic pregnancy (assessed at 18-month follow-up) was defined as a current or past 6-month pregnancy with intent to delay pregnancy for more than 1 year at 2020 survey.

**Results** While fertility intentions remained stable, contraceptive dynamics varied by gender—young men both adopted and discontinued coital-dependent methods, whereas young women adopted coital-dependent or short-acting methods at 12-month follow-up (2020). Current pregnancy was highest at 2020 (4.8%), and approximately 2% at 2019 and 2021. Unintended pandemic pregnancy prevalence was 6.1%, with increased odds for young women recently married (adjusted OR (aOR)=3.79; 95% confidence interval (CI) 1.83–7.86); recent contraceptive use was protective against unintended pandemic pregnancy (aOR=0.23; 95% CI 0.11–0.47).

**Conclusions** Current pregnancy in Nairobi was highest at the height of the COVID-19 pandemic (2020), and subsided to pre-pandemic levels by 2021 data collection; however, requires further monitoring. New marriages posed considerable risk for unintended pandemic pregnancy.

## STRENGTHS AND LIMITATIONS OF THIS STUDY

⇒ The COVID-19 pandemic raised concerns for unintended pregnancy for adolescent girls and young women (AGYW); however, until now, data on this distal outcome had not been available due to time lag.

⇒ To our knowledge, this is the first study to longitudinally examine unintended pregnancy incurred during the COVID-19 pandemic among AGYW in sub-Saharan Africa; limitations of retrospective unintended pregnancy assessment are well established, making prospective cohort data uniquely valuable to address this topic.

⇒ The assessment of unintended pandemic pregnancy, while our best estimate, may be subject to measurement error given that data collection was not continually ongoing throughout the pandemic.

⇒ Data were collected immediately prior to the pandemic and at two points after the implementation of COVID-19 restrictions in Nairobi—as Kenya continues to experience COVID-19 infection waves and reinstates mobility restrictions to curb infection spread, ongoing data collection is needed to understand impacts.

⇒ Nairobi is a major urban centre and results should not be generalised to all parts of Kenya, where AGYW may face different challenges surrounding sexual and reproductive health.

Contraceptive use remains a crucial preventive strategy to averting unintended pregnancy, particularly for married young women.

## INTRODUCTION

Adolescent pregnancy is a human rights concern with gendered repercussions that disproportionately impact adolescent girls and young women (AGYW).[1][2] Both intended and unintended pregnancies can alter AGYW's life trajectories, including by decreasing their ability to meet educational,

business/financial and family goals.[1] Further, early pregnancy can have negative health effects—adolescents are at risk for increased birth complications, including obstructed labour, fistula, premature delivery, low birth weight, pre-eclampsia/eclampsia and maternal and perinatal mortality, compared with their older counterparts.[1 3] While complete adolescent pregnancy data are difficult to obtain, largely due to under-reporting and stigmatisation of unintended pregnancies that end in abortions, the evidence is clear that the majority of the global adolescent pregnancy burden is concentrated within sub-Saharan Africa, where adolescent pregnancy rates in most countries exceed 100 per 1000 women aged 15–19.[4]

Motivations and intentions surrounding adolescent pregnancy are complex and require a socioecological lens.[5] Pregnancy intendedness varies substantially by setting and in many contexts, cultural and familial norms may promote early childbearing in lieu of economic and educational attainment.[2] Factors linked to adolescent pregnancy in sub-Saharan Africa include low education, lack of familial education, poverty and limited communication/knowledge surrounding sexual and reproductive health (SRH).[6 7] Ecological data further correlate adolescent pregnancy with high gender inequality at the national level.[8] While rural adolescents are generally at heightened risk of pregnancy, largely due to decreased access to SRH services,[6 7] urban adolescents have long been a target group for SRH programmes given the population's growing size and unique needs.[9] Many AGYW migrate to cities to seek out financial opportunities, but instead are met with stigma and shame when seeking healthcare services[10 11]; these barriers may be more pronounced for unmarried youth given social norms dictating sex as a means for procreation and stigmatisation of sexual activity outside of marriage, and in turn, increase unmet SRH needs.[12]

When the COVID-19 outbreak was declared a pandemic by WHO in March 2020, the SRH community was concerned about the negative impact mitigation measures would have on women's SRH, and particularly the SRH of AGYW.[13–15] Early COVID-19 projections estimated that 15 million additional unintended pregnancies could occur over 1 year if COVID-related service disruptions affected 10% of women in need of SRH services in low and middle-income countries.[16] Unintended pregnancy is a distal outcome and requires time and prospective data to accurately assess; thus far, no studies have corroborated these pregnancy projections.[17 18] In lieu of unintended pregnancy data, contraceptive use is the most widely used proxy for unintended pregnancy—population-level data from four sub-Saharan African contexts indicate that contraceptive use among women in need increased across settings, contrary to warnings of unintended pregnancy increases.[19] While early trends are largely promising, some groups may be at increased risk of unintended pregnancy, namely adolescents and nulliparous women, due to increases in contraceptive need and reductions in contraceptive use during this period.[19]

Another study using facility data to assess facility admissions and antenatal, delivery and immunisation services reported slight reductions in services.[20] Adolescents may further face difficulties procuring contraception during the COVID-19 pandemic given limited household bargaining power and school closures.[21 22] Recent causal-comparative research from Western Kenya elucidates increased school dropout and pregnancy among specific to secondary school girls during COVID-19 compared with a pre-COVID-19 cohort.[23]

While multiple studies have examined youth contraceptive use and/or sexual activity in light of COVID-19,[17 18 21 22] no study to date has used prospective data to examine unintended pregnancy during the COVID-19 pandemic in sub-Saharan Africa for AGYW. Among urban youth in Nairobi, Kenya, this study aimed to (1) characterise fertility and contraceptive use dynamics by gender; (2) estimate pregnancy prevalence for AGYW over the course of the pandemic; and (3) assess factors associated with unintended pandemic pregnancy for young women.

## METHODS
### Study setting
This study takes place in urban Nairobi, Kenya, where adolescent pregnancy is a country priority[24]—per 2014 Demographic and Health Survey estimates, one-fifth of Kenyan AGYW aged 15–19 were pregnant or already mothers, and approximately half of unmarried, sexually active AGYW reported using a modern method of contraception.[25] The COVID-19 pandemic elevated mitigation of adolescent pregnancy within national policies. The first case of COVID-19 in Kenya was detected on 13 March 2020,[26] with school closures, national lockdown and mandatory curfew immediately following.[27] By late 2020 restrictions were eased; however, in summer 2021, case loads began rising again, prompting additional closures.[27] These restrictions, while essential to curbing the spread of COVID-19, were hypothesised to decrease access to essential health services, including SRH services.

### Study overview
This longitudinal study draws on data collected from the Nairobi Youth Respondent-Driven Sampling Survey (YRDSS), a cohort of adolescents and young adults in Nairobi collected at three timepoints prior to and during the COVID-19 pandemic: June to August 2019 (prepandemic), August to October 2020 (12-month follow-up) and April to May 2021 (18-month follow-up).

Briefly, the YRDSS began recruitment in June to August 2019 using respondent-driven sampling (RDS), a chain-based recruitment method that begins with purposefully selected seeds, followed by monitored peer-to-peer coupon distribution.[28] At the time of recruitment, eligible youth were aged 15–24 years, unmarried and residing in Nairobi for at least 1 year. The 2019 prepandemic survey round recruited 1357 participants, of whom 95% (1293/1357) consented for recontact and provided

contact information. This cohort was recontacted and surveyed in August to October 2020 (12-month follow-up; n=1217; 94% retention) and in April to May 2021 (18-month follow-up; n=1177; 97% retention).

Detailed study procedures are outlined elsewhere.[29]

## Data collection procedures

The prepandemic (2019) survey was specific to reproductive health needs and family planning behaviours for adolescents and young adults. Data collection occurred in person with surveys self-administered by participants or administered by trained resident enumerators in case of comprehension or technological difficulty. Data were entered onto tablets and uploaded to a cloud server using Open Data Kit (ODK) software. Surveys were conducted in English or Swahili based on participant preference.

Given the onset of the COVID-19 pandemic, follow-up surveys at 12 months (2020) and 18 months (2021) were designed specifically to examine the gendered impacts of COVID-19 on youth economic, health and social experiences, including pandemic impacts on youth contraceptive use and unintended pregnancy. In adherence to COVID-19 restrictions, interviews at these timepoints were conducted by the same resident enumerators via remote phone interviews, with responses recorded using the ODK software.

## Measures
### Outcomes

Primary outcomes comprised fertility and contraceptive use for both genders, as well as pregnancy for young women.

### Prospective fertility intentions

Prospective fertility intentions were assessed per gender at each of the three timepoints. Specifically, each participant was asked 'How long would you like to wait from now before the birth of a child?' with responses comprising 'x months', 'x years', 'would like to get pregnant soon/now', and 'can't get pregnant' (for women) or 'can't cause pregnancy' (for men), and 'do not want children'. For trend analysis, intentions were categorised as 'currently pregnant'; '<1 year'; '1–2 years'; '>2 years/other'.

### Current contraceptive use

Current contraceptive use was assessed per gender at each timepoint using standard items.[30] Contraceptive users were asked a current method, and if users reported more than one current method, they were then asked to identify the method they used 'mainly' or 'most of the time'. Responses were split into method effectiveness categories for trend analysis: (1) long-acting reversible contraception (LARC): intrauterine device and implant; (2) short acting: injectable and pills; (3) coital dependent: emergency contraception, male condoms and female condoms; (4) other/traditional: rhythm, withdrawal and herbal pill method.

### Period and current pregnancies

Period and current pregnancies were assessed per time-point for young women. At pre-pandemic baseline, young women were asked about ever pregnancy, whereas at 12-month and 18-month follow-ups, past 12-month and past 6-month pregnancies were assessed to capture between-survey outcomes. For all young women with a period pregnancy, current pregnancy is assessed via a single item, 'Are you currently pregnant?'. A dichotomous variable was then created for the primary outcome of interest, *unintended pandemic pregnancy*, defined as having a current pregnancy or past 6-month pregnancy at 18-month follow-up survey and wanting to wait more than 1 year for a pregnancy at last survey round (2020).

### Exposures for unintended pandemic pregnancy

All risk factors for unintended pandemic pregnancy were assessed at 12-month follow-up (2020) survey using standard measures,[29 30] and include age (15-19; 20-24); highest level of education completed (no school/primary/post-primary; secondary or higher); subjective household wealth tertile (low; middle; high); employment (someone else vs self); increased time with partner since COVID-19 restrictions (yes/no); ability to meet basic needs (yes/no); and contraceptive use at 2020 survey. Exposures were dichotomised based on underlying distributions to maximise power for multivariable analyses.

### Analytical sample

Within-timepoint analyses were restricted to participants with completed survey data for that round (n=605 young men, n=610 young women at prepandemic and 12-month follow-up, n=586 young men, n=591 young women at 18-month follow-up). Trend and prospective analyses were restricted to those with complete data at all three timepoints (n=586 young men, n=589 young women); for contraceptive dynamics, analyses were additionally restricted to those with a need for contraception (ie, sexually active, not pregnant (if female) and not wanting a child within the next year) at both the time of that survey and the prior survey round. A complete case approach was adopted given <1% missing outcome data (n=2 missing unintended pandemic pregnancies); sample size floats to accommodate small amounts of missing covariate data (<1%).

### Statistical analysis

Descriptive statistics examined the characteristics of youth at prepandemic, by gender, with design-based F-statistics classifying between-gender differences. Sankey diagrams depicted dynamics in fertility intentions and contraceptive use across the three survey rounds, by gender. Among youth who discontinued their method at 12-month or 18-month follow-up, the main reason for discontinuation was examined per gender and timepoint. For each round, period and current pregnancy prevalence was calculated for all young women, and among adolescent girls (<18 years old) and those wishing to avert

a pregnancy at last round (ie, did not want a child within the next year). Lastly, bivariate and multivariable logistic regressions examined the factors at 12-month follow-up (2020) associated with unintended pandemic pregnancy among young women via glm models with family binomial and link logistic to account for weighing and clustering. Factors with p values <0.1 from bivariate models were included within multivariable models, withstanding increased time with partner, omitted due to collinearity.

All analyses were conducted using Stata V.16 (College Station, Texas) with statistical significance set a priori at p<0.05. Sampling weights accommodate the RDS study design using RDS-II (Volz-Heckathorn) weights, postestimation adjustment based on 2014 Kenya Demographic and Health Survey population data (age, sex, education levels) and loss to follow-up. All estimates are weighted; statistical testing accounts for clustering among participants recruited by the same recruiter at baseline.

### Patient and public involvement

This community-engaged study sought public and end user input at all phases. During the formative research stage prior to the 2019 cohort recruitment, input from community-based, youth-serving organisations informed the study recruitment strategy for feasibility, survey measures and constructs to ensure relevance and study logistics to maximise participant comfort and confidentiality. All recruitment and procedures were conducted by trained resident enumerators selected from underlying communities, and who provided inputs on measures for clarity and aided in results interpretation. Findings were disseminated in November 2020 and again in September 2021 with stakeholders spanning policy sector, government representatives, elders/faith leaders, community-based organisations and youth leaders from the study communities.

### RESULTS

Characteristics of youth at prepandemic, by gender, are outlined in table 1. The majority of both young men and young women were aged 20–24 (60.7% and 57.2%, respectively) and had completed at least secondary education (62.5% and 57.2%, respectively). While roughly three of four youth had ever had sex, higher proportions of young men reported current involvement in a romantic relationship (young men: 66.5%; young women: 54.5%; p=0.04). Approximately 1 in 10 young men had a partner who had ever been pregnant and 8.0% of young women reported their own pregnancy experience. Future fertility intentions differed by gender, with higher proportions of young men reporting wanting a child soon/now (2.7% young men vs 0.2% young women) or within the next 2 years (15.8% young men vs 11.0% young women; p=0.05). Contraceptive use, method effectiveness and main current contraceptive method differed significantly by gender. Approximately three-quarters of young men reported current contraceptive use (75.2%), compared

with half of young women (54.7%; p<0.001). The majority (91.2%) of young men reported male condom use as their main contraceptive method, whereas young women used a range of methods (p<0.001).

Throughout the pandemic, young women's fertility intentions remained relatively stable (figure 1). Between prepandemic and 12-month follow-up (2020), largest shifts in fertility intention were seen among women who wanted to wait more than 2 years at prepandemic (88.6%; n=515/591) and then said they wanted a child in 1–2 years at 2020 (11.4%; n=80/591); this trend was similarly observed between 12-month and 18-month follow-ups, wherein 9.9% of all women (n=65/591) shifted from the greater than 2 years category to 1–2 years. The second largest shift was seen among young women who wanted to wait more than 2 years at prepandemic, but then reported pregnancy at 12-month (2020) follow-up (4.2%; n=17/591). From 12-month to 18-month follow-ups, a corresponding shift from pregnancy to wanting to wait more than 2 years was reported (4.8%; n=20/591).

Trends in fertility intentions for young men (figure 1) were similar; however, between 12-month (2020) and 18-month (2021) follow-ups, large portions of young men also sought to delay childbearing, with 6.8% changing from wanting a child within 1–2 years to wanting to wait 2 or more years (n=35/586).

Contraceptive dynamics throughout the pandemic showed substantial variation by gender (figure 2). Specifically, young men largely migrated between coital-dependent method use and non-use at both timepoints (including both adoption and discontinuation). Conversely, many young women began as non-users and adopted coital-dependent (33.5%; n=67/192) or short-acting methods (13.3%; n=30/192) at 12-month follow-up (2020). While some coital-dependent users at 12-month follow-up (2020) discontinued by 18-month follow-up (26.2%; n=28/113) (2021), some non-users also continued to take up coital-dependent (15.2%; n=25/120), short-acting (5.0%; n=6/120) and LARC methods (11.4%; 12/120).

At both 2020 and 2021 surveys and across genders, the primary reason for contraceptive discontinuation centred around partners, including fear of cheating, partner opposed, partner away, infrequent sex and not married (young women$_{2020}$: 64%; young women$_{2021}$: 62%; young men$_{2020}$: 66%; young men$_{2021}$: 76%; figure 3). Additionally, approximately one-fifth of both young women (21%) and young men (22%) at 2020 survey stated discontinuation for pregnancy-related reasons, namely wanting to become pregnant. COVID-related, access-related and personal concerns were less common.

Following trends in fertility intentions and contraceptive use, prevalence of pregnancy differed across survey rounds (table 2). Period prevalence ranged from 31.4% ever pregnant at prepandemic (2019) to 6.8% pregnant in the past 6 months at 18-month follow-up (2021). Current pregnancy was highest at the 12-month follow-up survey (2020) at 4.8%, whereas it hovered at approximately 2% at

**Table 1** Characteristics of youth at prepandemic (2019; n=1215), weighted

| | Young men (n=605) | Young women (n=610) | |
|---|---|---|---|
| | n (column %) | | P value* |
| **Demographic characteristics** | | | |
| Age | | | 0.38 |
| 15–19 | 238 (39.3) | 261 (42.8) | |
| 20–24 | 367 (60.7) | 349 (57.2) | |
| Highest level of education completed | | | 0.62 |
| Postprimary/primary or no school | 197 (32.6) | 211 (34.7) | |
| Secondary or higher | 408 (67.4) | 399 (65.4) | |
| **Relationship and reproductive characteristics** | | | |
| Ever had sex | 436 (72.0) | 441 (72.3) | 0.94 |
| Currently in a romantic/sexual relationship | 402 (66.5) | 332 (54.5) | **0.04** |
| Ever pregnant/partner has ever been pregnant | 62 (10.2) | 49 (8.0) | 0.35 |
| Future fertility intentions | | | **0.05** |
| Soon/now | 16.3 (2.7) | 1 (0.2) | |
| Within the next 2 years | 96 (15.8) | 67 (11.0) | |
| More than 2 years | 410 (67.7) | 446 (73.2) | |
| Does not want children or cannot have children | 7 (1.1) | 8 (1.4) | |
| Don't know | 76 (12.6) | 87 (14.2) | |
| **Contraceptive characteristics** | | | |
| Contraceptive use | 323 (75.2) | 249 (54.7) | **<0.001** |
| Method effectiveness† | | | **<0.001** |
| Long-acting reversible contraception | 3 (1.0) | 48 (19.5) | |
| Short acting | 2 (0.5) | 55 (22.5) | |
| Barrier or hormonal coital dependent | 286 (93.6) | 114 (46.5) | |
| Other/traditional | 15 (4.9) | 28 (11.4) | |
| Main contraceptive method† | | | **<0.001** |
| Implant | 3 (1.0) | 48 (19.1) | |
| Coil/IUD | 0 (0.0) | 7 (2.7) | |
| Injectable | 2 (0.5) | 43 (17.0) | |
| Contraceptive pill/oral contraceptives | 0 (0.0) | 12 (5.0) | |
| Emergency contraception | 6 (1.9) | 24 (9.7) | |
| Male condom | 281 (91.2) | 85 (34.0) | |
| Female condom | 0 (0.0) | 5 (1.8) | |
| Cycle beads/standard days method/rhythm | 7 (2.4) | 12 (4.6) | |
| Withdrawal | 8 (2.5) | 10 (3.9) | |
| Herbal pill method | 1 (0.5) | 5 (2.1) | |
| Place obtained main contraceptive method† | | | 0.28 |
| Hospital | 26 (9.1) | 34 (15.1) | |
| Health centre | 89 (30.1) | 69 (30.9) | |
| Clinic/other health provider | 26 (9.1) | 31 (13.7) | |
| Pharmacy | 91 (31.7) | 62 (27.6) | |
| Other | 55 (19.1) | 28 (12.7) | |

Method effectiveness: LARC=IUD, implant; short acting=injectable, pills; coital dependent=emergency contraception, male condoms, female condoms; other/traditional=rhythm, withdrawal, herbal pill method.
Bold indicates p<0.05.
*P value from design-based F-statistic.
†Among users of contraception.
IUD, intrauterine device; LARC, long-acting reversible contraception.

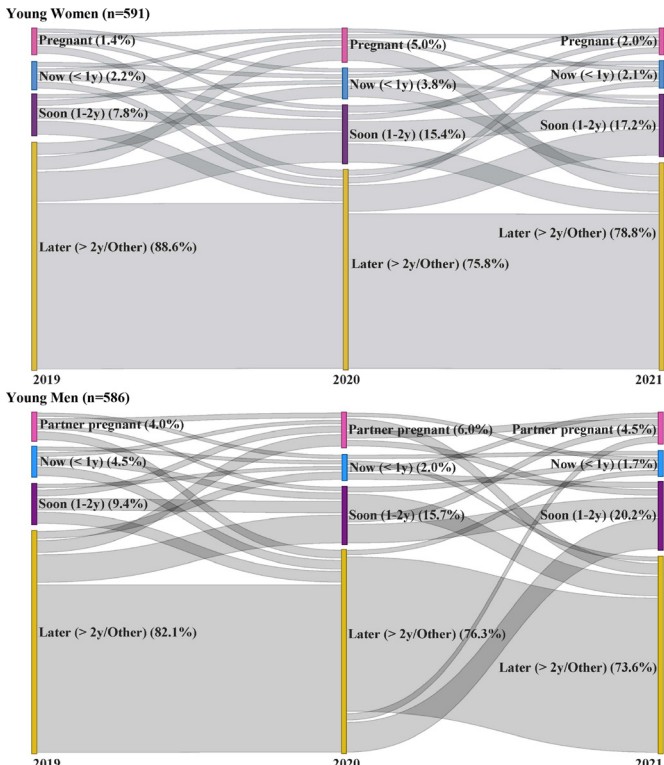

**Figure 1** Fertility intentions throughout the pandemic, by gender. Restricted to those with complete survey data at all three timepoints.

the 2019 and 2021 survey rounds. Unintended pandemic pregnancy (ie, current pregnancy or past 6-month pregnancy at the 18-month follow-up survey with no intention to get pregnant indicated at the previous survey round)

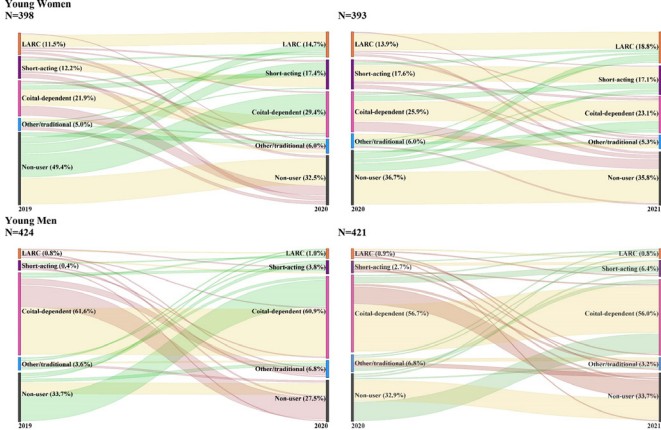

**Figure 2** Contraceptive dynamics throughout the pandemic by method effectiveness, by gender. Restricted to those who were sexually active and in need of contraception (not pregnant and did not want a child soon/now or within the next year) at both the time of that survey and the prior survey round, per gender. Method effectiveness: LARC=IUD, implant; short acting=injectable, pills; coital dependent=emergency contraception, male condoms, female condoms; other/traditional=rhythm, withdrawal, herbal pill method. IUD, intrauterine device; LARC, long-acting reversible contraception.

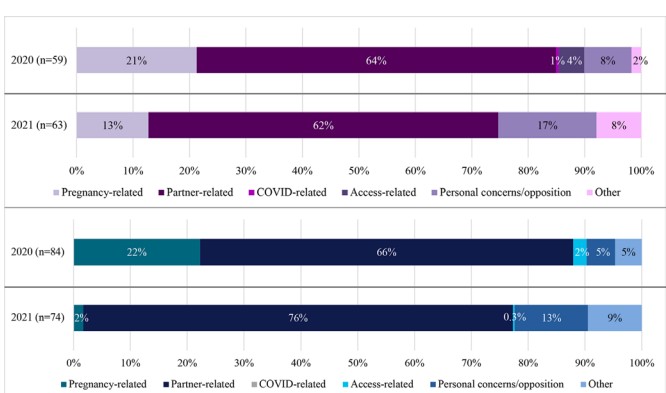

**Figure 3.** Main reason for discontinuation at 12-month (n=143) and 18-month (n=137) follow-ups. Pregnancy related: want to become pregnant. Partner related: fear of cheating, partner opposed, partner away, infrequent sex, not married. COVID related: could not go to facility due to COVID-19 restrictions. Access related: lack of access/too far, cost, preferred method not available, no method available, inconvenient to use, knows no alternate method. Personal concerns about/opposition to contraception: respondent opposed, interferes with body's processes, religious reasons, others opposed, fear of side effects, health concerns.

was 6.1%, with prevalence approximately half for girls less than age 18 (3.3%).

Table 3 presents the bivariate and multivariable factors at 12-month follow-up (2020) associated with unintended pandemic pregnancy. Recent marriage (at 2020 survey) was associated with a greater than four-fold increase in unintended pandemic pregnancy in both bivariate (OR=4.89; 95% CI 1.93–12.37; p<0.001) and multivariable models (adjusted OR (aOR)=4.16; 95% CI 1.65–10.51; p=0.006). Within the bivariate model, increased time with partner since COVID-19 restrictions was also associated with increased risk of unintended pandemic pregnancy (OR=3.69; 95% CI 1.61–8.46; p=0.002); however, it was excluded from multivariable models due to collinearity with marital status. Contraceptive use at 2020 survey contributed to an 80% reduction in unintended pandemic pregnancy (aOR=0.21; 95% CI 0.09–0.50; p<0.001). Additionally, within both bivariate models, young women with an inability to meet basic needs displayed a borderline protective effect against unintended pandemic pregnancy (OR=0.47; 95% CI 0.21–1.05; p=0.07); however, this association attenuated when adjusted.

## DISCUSSION

Within a narrative that has largely focused on shifting fertility intentions and contraceptive use amidst the COVID-19 pandemic,[16 19 31] these longitudinal results counter those for adult populations[17 18] and indicate that fears of unintended pregnancy for AGYW were indeed warranted. AGYW in Nairobi, Kenya, did not use effective methods of contraception and in turn experienced unintended pregnancy during the time of highest COVID-19 mitigation measures (2020 survey). Specifically, current

**Table 2** Prevalence of pregnancy across survey rounds, weighted

| | All AGYW | Adolescent girls (<18 years old at baseline) | Among AGYW wishing to avert a pregnancy at last round* |
|---|---|---|---|
| | % (n) | | |
| **Period prevalence** | | | |
| Pre-pandemic (2019) ever pregnant | 31.4 (191) | 6.2 (5) | – |
| 12-month follow-up (2020) past 12-month pregnancy | 12.5 (76) | 1.2 (1) | 11.2 (65) |
| 18-month follow-up (2021) past 6-month pregnancy | 6.8 (40) | 3.3 (1) | 6.6 (36) |
| **Current pregnancy** | | | |
| Pre-pandemic (2019) | 1.8 (11) | 0.0 (0) | – |
| 12-month follow-up (2020) | 4.8 (29) | 0.0 (0) | 4.6 (27) |
| 18-month follow-up (2021) | 2.0 (12) | 3.3 (1) | 1.6 (9) |
| Unintended pandemic pregnancy (current pregnancy or past 6-month pregnancy at 18-month follow-up, with no intention to get pregnant at last round (2020)) | 6.1 (36) | 3.3 (1) | – |

*Wished to wait for pregnancy >1 year at prepandemic (for 12-month follow-up assessment) and >6 months at 12-month follow-up (for 18-month follow-up assessment).
AGYW, adolescent girls and young women.

pregnancy more than doubled at the 2020 survey wave (4.8%) versus prepandemic (1.8%) or 2021 (2.0%) surveys, and the majority of pregnancies at 2020 survey (93%) were unintended. Counter to initial hypotheses, education and economic insecurity did not pose significant threats to unintended pandemic pregnancy—instead, risk among AGYW in Nairobi largely centred on newly initiated marriage.

Notably, recent marriage was the only risk factor associated with increased odds of unintended pandemic pregnancy within the multivariable models, where young women who recently married had a four-fold increase of unintended pandemic pregnancy, as compared with those who remained unmarried (p<0.001). Such findings bolster recent results from national cohorts of adult women predicting increases in unintended pregnancy for nulliparous and young women in Lagos, Nigeria[19]; the aforementioned study, however, did not highlight potential increases in unintended pregnancy for young women within Kenya.[19] Using longitudinal data from an existing cohort of this hard-to-reach population, our findings elucidate that it may not be age nor parity driving risk for AGYW, but rather power dynamics within a newly formed marriage. Pressures for newly married women may increase risk of unintended pregnancy, both during the COVID-19 pandemic and after, and require further research to understand the relationship dynamics for newly married couples. While partner pressures are most prominent, concurrent pressures from family and community members should not be downplayed,[32 33] and a socioecological framework should be adopted in future research to fully understand the risks of new marriage for AGYW's SRH. While programmes to increase age at marriage and keep girls in school are vital to ensure AGYW's autonomy in the decision to marry and their

future livelihoods, interventions to increase relationship quality and help young couples navigate the complexities of this new life stage must be simultaneously prioritised in order for young couples to prosper both individually and jointly.

Shifts in fertility intentions throughout the pandemic largely followed expected patterns and timelines (ie, women who wanted to wait more than 2 years at prepandemic stated that they then wanted a child within 1–2 years 1 year later). Some AGYW (4.2%; n=17), however, did report wanting to wait more than 2 years at prepandemic survey but then had an unintended pregnancy at 2020 survey—at 2021 survey, these young women again wanted to wait more than 2 years for their next pregnancy. While young men's fertility intentions followed similar patterns, between 12-month (2020) and 18-month (2021) follow-ups, 6.8% shifted from wanting a child within 1–2 years to wanting to wait 2 or more years—this shift in intentions could be in part due to the continued economic uncertainty and instability imposed by the COVID-19 pandemic. Results corroborate borderline significant results in multivariable models indicating that AGYW with an inability to meet their basic needs had a 50% reduction in unintended pandemic pregnancy. Ultimately, these results speak to the resilience of young people, as well as their initiative to prepare and financially plan for a pregnancy. Continued investment in youth economic empowerment can ensure young people are able to meet both their SRH and financial goals.

Moreover, results affirm the effectiveness of primary prevention for unintended pregnancy via contraceptive use. Contraceptive use at previous survey round was the only factor protective against unintended pandemic pregnancy—AGYW who used contraception, regardless of method, had an 80% reduction in unintended pandemic

**Table 3** Factors at 12-month follow-up (2020) associated with unintended pandemic pregnancy for young women (n=589)

| | Unintended pandemic pregnancy Row % | OR (95% CI) | aOR (95% CI) |
|---|---|---|---|
| **Factors assessed at 12-month follow-up (2020)** | | | |
| Age | | | |
| 15–19 | 6.5 | Ref | – |
| 20–24 | 5.9 | 0.90 (0.37, 2.19) | – |
| Highest level of education completed | | | |
| Postprimary/primary or no school | 6.9 | Ref | – |
| Secondary or higher | 5.7 | 0.82 (0.32, 2.08) | – |
| Household wealth | | | |
| Low | 6.4 | Ref | – |
| Middle | 5.6 | 0.87 (0.33, 2.34) | – |
| High | 6.2 | 0.96 (0.34, 2.70) | – |
| Married | | | |
| No | 4.5 | Ref | Ref |
| Yes | 18.9 | **4.89 (1.93, 12.37)*** | **4.16 (1.65, 10.51)** |
| Employed | | | |
| Student, caregiver, unemployed | 5.1 | Ref | – |
| Employed formally or informally | 7.2 | 1.45 (0.60, 3.49) | – |
| Prime earner in household | | | |
| Someone else | 5.9 | Ref | – |
| Self | 7.4 | 1.28 (0.46, 3.57) | – |
| Increased time with partner since COVID-19 restrictions | | | |
| No | 3.8 | Ref | – |
| Yes | 12.8 | **3.69 (1.61, 8.46)** | – |
| Ability to meet basic needs | | | |
| No | 4.1 | 0.47 (0.21, 1.05)† | 0.54 (0.23, 1.29)† |
| Yes | 8.4 | Ref | Ref |
| Contraceptive use | | | |
| No | 14.4 | Ref | Ref |
| Yes | 3.7 | **0.23 (0.10, 0.53)*** | **0.21 (0.09, 0.50)*** |

Increased time with partner omitted from multivariable model due to collinearity.
Bold indicates p<0.05.
*p<0.05; **p<0.01; ***p<0.001
†p<0.10
aOR, adjusted OR.

pregnancy, compared with those who were not using contraception at the last survey round. Youth must have continued access to non-stigmatising, youth-friendly and low-cost SRH services. For AGYW who experience unintended pregnancies, during the COVID-19 pandemic and otherwise, access to postpartum contraception is imperative to deter against rapid repeat pregnancy, while ensuring that timing of future pregnancies aligns with a woman's reproductive intentions.

While early research projected contraceptive use decreases due to COVID-related reasons,[16] urban AGYW in Nairobi did not cite inability to go to the facility due to COVID-19 restrictions as their primary reason for discontinuation. At both 2020 and 2021 and across genders, the primary reason for contraceptive discontinuation centred around partners (fear of cheating, partner opposed, partner away, infrequent sex, not married). Notably, the included survey item for COVID-related reasons only measured access challenges via inability to visit facilities due to restrictions. Moreover, respondents were only instructed to select their *primary* reason for discontinuation, though multiple reasons could have been experienced. Recent COVID-19 research has reported negative impacts of COVID-19 on youth

relationships,[21][22][34] including dissolution of partnerships, increased economic dependency, intimate partner violence and accelerated marriage timelines; altogether, the current research base describes the difficulty of disentangling reasons for discontinuation, with partnership dynamics potentially serving as a proxy for more distal COVID-related reasons. Partnership and COVID factors should continue to concurrently be measured in survey research, with further longitudinal research needed to disentangle their full and partial effects on SRH outcomes.

Results should be interpreted considering study limitations. Foremost, data were collected immediately prior to the pandemic and at two points after the implementation of COVID-19 restrictions in Nairobi—as Kenya continues to experience COVID-19 infection waves and reinstates mobility restrictions to curb infection spread, ongoing data collection is needed to understand impacts on AGYW's SRH and well-being. Moreover, the assessment of unintended pandemic pregnancy, while our best estimate, may be subject to measurement error given that data collection was not continually ongoing throughout the pandemic. Namely, some AGYW who were pregnant at the 2020 survey round may have indeed experienced an unintended pandemic pregnancy, although we used the 18-month (2021) survey for more accurate estimates. We further recognise that fertility intentions are dynamic; while we used prospective intention measures from the most recent previous timepoint to minimise recall biases associated with intention measurement, we recognise that fertility intentions could shift throughout our short follow-up window. Period pregnancy prevalence estimates use different referent timepoints and therefore are not comparable; as such, significance testing between times was not used and these data should be interpreted as purely descriptive. Nairobi is a major urban centre and results should not be generalised to all parts of Kenya, where AGYW may face different challenges surrounding SRH.

As of February 2023, Kenya has experienced seven COVID-19 infection waves, and the health and economic impacts of these waves may not yet to be fully uncovered; recent notable policy shifts, however, signal positive steps for youth in the country.[35] Growing momentum seeks to advance women's safety and reproductive rights, including via affirming access to abortion services, and advocating for the elimination of gender-based violence as a national priority. These prospective data, however, highlight that continued policies are needed to affirm reproductive rights and access to quality contraceptive and SRH care for adolescents and young adults. Foremost, continuous access to quality health education, youth-friendly contraceptive services and universal healthcare more broadly is imperative to protect the reproductive autonomy of both young men and young women. Further evidence-informed policy efforts should centre on dismantling fertility pressures within the community and partner dyad, particularly for AGYW transitioning from dating to marital relationships and still seeking to avert pregnancy.

Additional data collection in this arena can aid in monitoring fertility pressures, contraceptive access and use and quality of care to better understand the continued SRH needs of this vulnerable group. Such youth-friendly policies are crucial for advancing adolescent and young adult SRH both in times of global public health crises and in times of relative stability.

**Acknowledgements** We acknowledge Dr Caroline Moreau for her assistance in defining our analytical sample and outcome variables.

**Contributors** SNW contributed to study design, literature search, data analysis, data interpretation and writing. MEB contributed to study design, literature search, data analysis, figures, data interpretation and writing. MT contributed to study design, data interpretation and writing. BD contributed to study design, data interpretation and writing. GW-N contributed to data interpretation and writing. MRD contributed to study design, data interpretation and writing, and served as the guarantor of the study. PG contributed to study design, data collection, data interpretation and writing. All authors read and approved the final version of this manuscript.

**Funding** This work was supported, in whole, by the Bill & Melinda Gates Foundation (010481).

**Disclaimer** The funders had no role in the study design, collection, analysis, and interpretation of data, in writing of the report, or in the decision to submit for publication.

**Competing interests** None declared.

**Patient and public involvement** Patients and/or the public were involved in the design, or conduct, or reporting, or dissemination plans of this research. Refer to the Methods section for further details.

**Patient consent for publication** Not applicable.

**Ethics approval** This study involves human participants and ethical approval was obtained from the Ethics Review Committee at Kenyatta National Hospital/University of Nairobi (P294/04/2019) and the Institutional Review Board at Johns Hopkins Bloomberg School of Public Health (00009496). Participants gave informed consent to participate in the study before taking part.

**Provenance and peer review** Not commissioned; externally peer reviewed.

**Data availability statement** Data are available upon reasonable request. Data are available upon request from www.pmadata.org.

**ORCID iD**
Shannon N Wood http://orcid.org/0000-0003-4389-3526

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
