## [Reviewer comments · BMJ Open]

ARTICLE DETAILS

TITLE (PROVISIONAL)	Fertility and Contraceptive Dynamics amidst COVID-19: Who is at Greatest Risk for Unintended Pregnancy among a Cohort of Adolescents and Young Adults in Nairobi, Kenya?
AUTHORS	Wood, Shannon; Byrne, Meagan E.; Thiongo, Mary; Devoto, Bianca; Wamue-Ngare, G; Decker, Michele; Gichangi, Peter

VERSION 1 – REVIEW

REVIEWER	Backhaus , Andreas Bundesinstitut fur Bevolkerungsforschung
REVIEW RETURNED	24-Nov-2022

GENERAL COMMENTS	Please find below a number of comments and suggestions from my review of your manuscript “Fertility and Contraceptive Dynamics amidst COVID-19: Who is at Greatest Risk for Unintended Pregnancy among Adolescents and Young Adults in Nairobi, Kenya”. Terminology: In your Key Messages and in the Introduction, you state that no study so far has used prospective data to “predict unintended pregnancy” during COVID-19. In my understanding of the term, a prediction would involve a forecast of future unintended pregnancies. However, this is not what your study does, as you present pregnancy rates in different past periods and a regression analysis of factors associated with unintended pregnancies, both based on existing data. You may consider omitting the usage of this term. If my understanding is correct, you define an unintended pregnancy as a pregnancy that has occurred in a given survey round to a woman or a man’s partner who had in a previous survey round declared the intention to delay fertility for a longer period than the period that has elapsed between these two consecutive survey rounds, i.e. a pregnancy that has occurred earlier than the past intentions would suggest. However, it seems entirely possible that an individual’s fertility intentions had changed in favor of an earlier pregnancy between survey rounds, with the changed intentions then having prompted a pregnancy. In this case, the timing of the pregnancy would still contradict the initially stated fertility preference, but it would not be an unintended pregnancy due to the altered preference. The manuscript does not appear to discuss this possibility, which however seems crucial for a clear delineation of your main outcome. It might further cast doubt on your finding that 93% of the pregnancies in 2020 were unintended, which seems like an extremely high rate. Related literature:
--

	Regarding the existing literature on contraceptive use and/or sexual activity during COVID-19, I may point to Backhaus (2022) in addition to Karp et al. (2021) and Decker et al. (2021); the former finds no significant change in pregnancies within younger age groups during COVID-19 compared to pre-pandemic levels in Kenya but significantly higher rates of modern contraceptive usage: Backhaus, A. (2022). Pregnancies and contraceptive use in four African countries during the COVID-19 pandemic. Vienna Yearbook of Population Research, 20(1). DOI: https://doi.org/10.1553/populationyearbook2022.dat.4 Study design: The months of the respective year during which a survey wave was collected change with each wave, from June-August to August-October to April-May. Births and hence pregnancies follow certain seasonal patterns across the year though. Currently, your manuscript does not discuss the potential influence that these patterns may exert on the frequency of pregnancy in the different waves and how you could curtail the implied biases. Results and Discussion: Table 2 is central to your results, as it reports the prevalence of pregnancy in the different survey periods. However, what is missing are statistical tests of the various rates in order to assess whether they are actually statistically different from each other. Further, I wonder what we can learn from the presented unintended pandemic pregnancy rate, as the table does not present an unintended pre-pandemic pregnancy rate; hence, it is actually not possible to judge whether unintended pregnancies have increased or decreased during the pandemic. When presenting the results from Table 3, the manuscript reads “young women with an inability to meet basic needs displayed a borderline protective effect against unintended pandemic pregnancy”. Besides this result being quite counter-intuitive, the table actually states that women with an ability to meet basic needs exhibited an unintended pandemic pregnancy rate of 4.1%, compared to 8.4% among women that did not have the ability to meet basic needs. I suggest validating the displayed rates and estimates in the table, as well as the corresponding text. In your Discussion, you state: “Results on fertility intentions, contraceptive use, and pregnancy all tell the same story – AGYW decreased their contraceptive use and in turn, experienced unintended pregnancy, (...)”. However, I do not see this statement supported by the evidence presented – at least, in Figure 2, the share of non-users of contraceptives declines substantially among both women and men. As you show in Figure 3, the share of individuals who state they have discontinued contraceptive usage for COVID-related reasons is minuscule. This finding appears to suggest that it is unlikely that the pandemic has notably affected pregnancies via a discontinuation of contraceptive usage, but a corresponding discussion is missing from the manuscript. While you discuss the positive association between unintended pregnancy and recent marriage in detail, an important yet missing aspect in the context of your theme would be whether this association could have been moderated by COVID-19.
--	---

REVIEWER	Terefe Tucho, Gudina Jimma University
REVIEW RETURNED	19-Dec-2022

GENERAL COMMENTS	The aim of the study was to characterize fertility and contraceptive use dynamics by gender; estimate pregnancy prevalence over the pandemic and assess factors associated with unintended pandemic pregnancy for young women in Nairobi Kenya. The finding provides an insight on the reproductive impacts of public health emergency that requires routine and continuous follow up. The following are my specific comments.  1. What are the policy implications of the results? 2. What do you recommend to overcome the unexpected problems in the future public health emergencies?
--

VERSION 1 – AUTHOR RESPONSE

Response to Reviewer 1: Dr. Andreas Backhaus

Terminology:

1. In your Key Messages and in the Introduction, you state that no study so far has used prospective data to “predict unintended pregnancy” during COVID-19. In my understanding of the term, a prediction would involve a forecast of future unintended pregnancies. However, this is not what your study does, as you present pregnancy rates in different past periods and a regression analysis of factors associated with unintended pregnancies, both based on existing data. You may consider omitting the usage of this term.

Authors’ Response: We have edited this term accordingly (pg. 4, line 42).

2. If my understanding is correct, you define an unintended pregnancy as a pregnancy that has occurred in a given survey round to a woman or a man’s partner who had in a previous survey round declared the intention to delay fertility for a longer period than the period that has elapsed between these two consecutive survey rounds, i.e. a pregnancy that has occurred earlier than the past intentions would suggest. However, it seems entirely possible that an individual’s fertility intentions had changed in favor of an earlier pregnancy between survey rounds, with the changed intentions then having prompted a pregnancy. In this case, the timing of the pregnancy would still contradict the initially stated fertility preference, but it would not be an unintended pregnancy due to the altered preference. The manuscript does not appear to discuss this possibility, which however seems crucial for a clear delineation of your main outcome. It might further cast doubt on your finding that 93% of pregnancies in 2020 were unintended, which seems like an extremely high rate.

Authors’ Response: Thank you for your critical feedback on this measure. While we agree that pregnancy intentions are not static, we clarify that, to date, prospective intentions are the most valid measure for assessing unintended pregnancy given the social desirability biases that occur with retrospective intention measurement. We further clarify that there were only six months between 12-month and 18-month follow-up, which were the data points used to assess unintended pandemic pregnancy. We have added details to the limitations section to discuss potential biases with this measure (pg. 11, lines 29-31), and expanded on our discussion surrounding continuous data collection. Figure 1 confirms the relative stability of pregnancy intentions throughout the data collection time points.

We further contextualize the finding that 93% of pregnancies in 2020 were unintended with details on our sample. These are unmarried or very recently married young women aged 16-25 at the onset of a pandemic, who are likely facing unprecedented economic and social uncertainty. Therefore, we do not find this proportion to be suspiciously high.

Related literature:

3. Regarding the existing literature on contraceptive use and/or sexual activity during COVID-19, I may point to Backhaus (2022) in addition to Karp et al. (2021) and Decker et al. (2021); the former finds no significant change in pregnancies within younger age groups during COVID-19 compared to pre-pandemic levels in Kenya but significantly higher rates of modern contraceptive usage:

Backhaus, A. (2022). Pregnancies and contraceptive use in four African countries during the COVID-19 pandemic. *Vienna Yearbook of Population Research*, 20(1).

DOI: <https://doi.org/10.1553/populationyearbook2022.dat.4>

Authors' Response: We have added this article, as well as a more recent article using PMA data from Moreau et. al.

Study design:

4. The months of the respective year during which a survey wave was collected change with each wave, from June-August to August-October to April-May. Births and hence pregnancies follow certain seasonal patterns across the year though. Currently, your manuscript does not discuss the potential influence that these patterns may exert on the frequency of pregnancy in the different waves and how you could curtail the implied biases.

Authors' Response: Thank you for this suggestion. We clarify that we are examining both period pregnancy and current pregnancy to examine pregnancy patterns in a number of different ways. Moreso than seasonality, trends in the COVID-19 pandemic and lockdown measures (i.e., school openings) could potentially drive these changes—therefore, we felt it was important to present the findings via numerous metrics.

Results and Discussion:

5. Table 2 is central to your results, as it reports the prevalence of pregnancy in the different survey periods. However, what is missing are statistical tests of the various rates in order to assess whether they are actually statistically different from each other. Further, I wonder what we can learn from the presented unintended pandemic pregnancy rate, as the table does not present an unintended pre-pandemic pregnancy rate; hence, it is actually not possible to judge whether unintended pregnancies have increased or decreased during the pandemic.

Authors' Response: Thank you for your attention to this table. We fully agree that not having a pre-pandemic unintended pregnancy rate is a limitation—we are clear in the table that our time periods slightly differ. Therefore, we also feel that it would be inappropriate to test for significance between these periods and present prevalence as purely descriptive. We have ensured throughout that we are not falsely implying increases/decreases in unintended pregnancy during the pandemic. We have also added this sentence to the limitations section: “Period pregnancy prevalence estimates utilize different referent timepoints and therefore are not comparable; as such, significance testing between times was not utilized and these data should be interpreted as purely descriptive” (pg. 11, lines 31-33).

VERSION 2 – REVIEW

REVIEWER	Backhaus , Andreas Bundesinstitut fur Bevölkerungsforschung
REVIEW RETURNED	13-Mar-2023
GENERAL COMMENTS	Thank you for responding to the points raised in my previous report. While I think the manuscript has improved, a number of problematic points remain from my point of view. 1. In my previous report, I raised concerns that your repeated measurements of fertility, contraceptive use and pregnancy might be affected by factors such as seasonality and that an informative estimation of pregnancy prevalence would require attempts to adjust for such factors before performing a statistical comparison. In your

	replies to points 4.) and 5.) in the previous report, you appear to acknowledge these limitations and explain why you refrain from significance testing. However, this leads back to my question what actually can then be learned from the presented pregnancy rates if they are not comparable with each other for a number of reasons. Note that not only the period pregnancy rate but also the current pregnancy rate can be affected by the varying temporal distance between the various survey rounds. 2. Further, in your response to point 5.) in my previous report, you state that "We have ensured throughout that we are not falsely implying increases/decreases in unintended pregnancy during the pandemic." This appears to be contradicted by your Discussion, in which you claim that "fears of unintended pregnancy for AGYW were indeed warranted. [...] Specifically, current pregnancy more than doubled at the 2020 survey wave (4.8%) versus pre-pandemic (1.8%) or 2021 (2.0%) surveys" - implying increases/decreases in unintended pregnancies based on rates that are affected with problems of comparability and have not been tested for their statistical difference. 3. Regarding point 6.) in the previous report, after correcting the typo, the result still implies that women who are not able to meet basic needs are less(!) likely to have an unintended pregnancy than women who are able to meet basic needs - which contradicts the intuition that women who cannot meet basic needs should be more vulnerable to unintended pregnancies due to a presumed lack of access to contraceptives and health centers, etc. I may suggest to double-check this result and, if found correct, to provide a brief intuition. 4. Regarding point 7.) in the previous report, I still cannot agree that your results "all tell the same story" - while your regression clearly shows a negative association between unintended pregnancy and contraceptive use, there was not, if I understand the paragraph correctly, a crisis of contraceptive usage but rather the opposite, as the share of non-users has declined. If you are implying that the coital-dependent and short-acting methods adopted by previously non-using young women are not effective methods of contraception, I do not see the reason why. 5. In the Abstract and throughout the manuscript, you describe recent marriages as a risk factor for unintended pregnancy ("New marriages posed considerable risk for unintended pandemic pregnancy."). Given that you estimate only an association between unintended pregnancy and recent marriage, I suggest refraining from this characterization. While the mechanisms that you describe could well be active in this case, the causality could easily run both ways: Recent marriages being formed only because of a preceding unintended pregnancy, in which case a marriage would not be a risk for but a consequence of an unintended pregnancy. 6. Considering the timing of the survey rounds, I may note that the individuals surveyed in June 2019 were revisited in August 2020 at the earliest, resulting in a follow-up period of 14 instead of 12 months. Similarly, the third survey round starting in April 2021 would imply a follow-up period of 22 instead of 18 months.
--	---

REVIEWER	Terefe Tucho, Gudina Jimma University
-----------------	--

REVIEW RETURNED	08-Mar-2023
GENERAL COMMENTS	thank you for addressing all the comments.

VERSION 2 – AUTHOR RESPONSE

Response to Reviewer 1: Dr. Andreas Backhaus, Bundesinstitut für Bevölkerungsforschung

Thank you for responding to the points raised in my previous report. While I think the manuscript has improved, a number of problematic points remain from my point of view.

Authors' Response: Thank you for your continued review of our manuscript.

1. In my previous report, I raised concerns that your repeated measurements of fertility, contraceptive use and pregnancy might be affected by factors such as seasonality and that an informative estimation of pregnancy prevalence would require attempts to adjust for such factors before performing a statistical comparison. In your replies to points 4.) and 5.) in the previous report, you appear to acknowledge these limitations and explain why you refrain from significance testing. However, this leads back to my question what actually can then be learned from the presented pregnancy rates if they are not comparable with each other for a number of reasons. Note that not only the period pregnancy rate but also the current pregnancy rate can be affected by the varying temporal distance between the various survey rounds.

Authors' Response: Many thanks for your continued concerns. Per the Editorial Review, we note that the described concerns are inherent limitations of observational studies. Despite these limitations, described on pg. 11, lines 31-33, we believe that our multiple measurements provide a thorough, descriptive picture of adolescent and young adult pregnancy in Nairobi throughout the COVID-19 pandemic.

2. Further, in your response to point 5.) in my previous report, you state that "We have ensured throughout that we are not falsely implying increases/decreases in unintended pregnancy during the pandemic." This appears to be contradicted by your Discussion, in which you claim that "fears of unintended pregnancy for AGYW were indeed warranted. [...] Specifically, current pregnancy more than doubled at the 2020 survey wave (4.8%) versus pre-pandemic (1.8%) or 2021 (2.0%) surveys" - implying increases/decreases in unintended pregnancies based on rates that are affected with problems of comparability and have not been tested for their statistical difference.

Authors' Response: Thank you for this point. Point 5 in your previous report was specifically detailing period prevalence and discussing not having a pre-pandemic baseline comparison. The current text in the discussion focuses on comparison of current pregnancy prevalence throughout the pandemic—these differences are not stated as significantly different because as you note, statistical differences were not run.

3. Regarding point 6.) in the previous report, after correcting the typo, the result still implies that women who are not able to meet basic needs are less(!) likely to have an unintended pregnancy than women who are able to meet basic needs - which contradicts the intuition that women who cannot meet basic needs should be more vulnerable to unintended pregnancies due to a presumed lack of access to contraceptives and health centers, etc. I may suggest to double-check this result and, if found correct, to provide a brief intuition.

Authors' Response: We have confirmed these findings and point this review to pg. 10, lines 35-41 where these results are discussed.

4. Regarding point 7.) in the previous report, I still cannot agree that your results "all tell the same story" - while your regression clearly shows a negative association between unintended pregnancy and contraceptive use, there was not, if I understand the paragraph correctly, a crisis of contraceptive usage but rather the opposite, as the share of non-users has declined. If you are implying that the coital-dependent and short-acting methods adopted by previously non-using young women are not effective methods of contraception, I do not see the reason why.

Authors' Response: We have edited this line to increase clarity (pg. 10, line 7).

5. In the Abstract and throughout the manuscript, you describe recent marriages as a risk factor for unintended pregnancy ("New marriages posed considerable risk for unintended pandemic pregnancy."). Given that you estimate only an association between unintended pregnancy and recent marriage, I suggest refraining from this characterization. While the mechanisms that you describe could well be active in this case, the causality could easily run both ways: Recent marriages being formed only because of a preceding unintended pregnancy, in which case a marriage would not be a risk for but a consequence of an unintended pregnancy.

Authors' Response: We clarify that we are specifically looking at unintended pandemic pregnancy (i.e., defined as having a current pregnancy or past 6-month pregnancy at 18-month follow-up survey and wanting to wait more than one year for a pregnancy at last survey round (2020)). Given the longitudinal nature of the design, recent marriage was assessed at 2020 and precedes the unintended pandemic pregnancy.

6. Considering the timing of the survey rounds, I may note that the individuals surveyed in June 2019 were revisited in August 2020 at the earliest, resulting in a follow-up period of 14 instead of 12 months. Similarly, the third survey round starting in April 2021 would imply a follow-up period of 22 instead of 18 months.

Authors' Response: We clarify that these were the minimal possible times between surveys.

Response to Reviewer 2 Dr. Gudina Terefe Tucho, Jimma University

Thank you for addressing all the comments.

Authors' Response: Many thanks for your thorough continued review of our manuscript.